

# Advancing Ecohydrological Modelling: Coupling LPJ-GUESS with ParFlow for Integrated Vegetation and Surface-Subsurface Hydrology Simulations

Zitong Jia[1], Shouzhi Chen[1], Yongshuo H. Fu[1,6,*], David Martín Belda[5], David Wårlind[4], Stefan Olin[4], Chongyu Xu[3], Jing Tang[2,4]

[1]College of Water Sciences, Beijing Normal University, Beijing, 100875, China
[2] Center for Volatile Interactions (VOLT), Department of Biology, Universitetsparken 15, Copenhagen, 2100, Denmark
[3]Department of Geosciences, University of Oslo, Oslo N-0316, Norway
[4]Department of Physical Geography and Ecosystem Science, University of Lund, Lund, 223 62, Sweden
[5]Institute of Meteorology and Climate Research Atmospheric Environmental Research (IMK-IFU), Karlsruhe Institute of Technology (KIT), Garmisch-Partenkirchen, 82467, Germany
[6]Plants and Ecosystems, Department of Biology, University of Antwerp, Antwerp, Belgium

*Correspondence to*: Yongshuo H. Fu (yfu@bnu.edu.cn)

**Abstract.** Climate change accelerates the global hydrological cycle, which has escalating impacts on human health and the socioeconomic development. However, many existing Earth system models neglect the more complex processes of topography-driven vegetation-surface-groundwater interactions, thereby failing to accurately capture climate-hydrological responses. To address this gap, we integrate the three-dimensional surface-subsurface hydrological model ParFlow with the dynamic global vegetation model LPJ-GUESS to investigate how lateral groundwater flow and vegetation dynamics jointly regulate hydrological fluxes. The fully coupled ParFlow-LPJ-GUESS (PF-LPJG) model and stand-alone LPJ-GUESS model were used to run hydrological simulations at a resolution of 10km across the Danube River Basin. A comprehensive evaluation of multiple hydrologic variables - including streamflow, surface soil moisture (SM), evapotranspiration (ET), and water table depth (WTD) was conducted using in situ and remote sensing (RS) observations based on a 38-year (1980-2018) model simulation. The results demonstrate that the PF-LPJG model substantially improves streamflow and surface soil moisture simulations without requiring parameter calibration compared to stand-alone LPJ-GUESS, mitigates the underestimation of summer low flows during dry years, increases the accuracy of peak flow timing in wet years, and achieves a Kling-Gupta Efficiency (KGE) > 0.5 and Spearman's ρ > 0.80 at over 80 % of gauging stations. Seasonal soil moisture anomalies are better captured (R = 0.51) compared to satellite-based products. Additionally, the modelled WTD agrees well with in-situ monitoring-well data, as indicated by a low RSR value (~1.31, Root Mean Square Error-observations Standard deviation Ratio). Notably, the coupled model improves the representation of bare-soil evaporation and reduces transpiration-to-evaporation (T/E) ratio fluctuations, aligning more closely with the GLEAM v4.2 product. The coupled model PF-LPJG entails a mechanistic framework for capturing bidirectional interactions among surface-subsurface water, vegetation dynamics and ecosystem biogeochemical processes, which can be applied to other catchments or climatic conditions to deeply analyze climate-induced modification on vegetation-water-carbon interactions.



## 1 Introduction

Global climate change is accelerating the hydrological cycle and intensifying its impacts on ecosystems and socio-economic systems (Alfieri et al., 2017, Geng et al., 2020). Groundwater attenuates climate variations through its processes of recharge, subsurface storage, and discharge (van Tiel, 2024; de Graaf et al., 2019). It plays a vital role in ecosystems by significantly shaping recharge patterns and water resource distribution, while ensuring stable water supply for both ecological systems and human demands (Famiglietti, 2014). Concurrently, extensive literature highlights that vegetation dynamics exert significant

feedback on the land-atmosphere system, affecting energy balance, water fluxes, and regional climate (Lian et al., 2018; Forzieri et al., 2020; Piao et al., 2019; Alkama et al., 2016). As the Earth system faces unprecedented climate alterations and anthropogenic pressures, the roles of groundwater and vegetation processes in the global hydrological cycle become increasingly pivotal (Yang et al., 2023; Wang et al., 2024). However, gaining a comprehensive understanding of the bidirectional interactions between vegetation dynamics and groundwater, as well as their response to climate change, remains

a significant challenge.

Existing studies have revealed a strong link between water table depth and land-atmosphere energy fluxes. Condon et al. (2013) demonstrated a consistent correlation between water table depth and latent heat flux within specific regions. The interaction between surface water and plant-available groundwater enhances the control of latent heat fluxes and transpiration partitioning (Good et al., 2015). Globally, plant transpiration accounts for approximately 64 ± 13% of total evapotranspiration, with 65 ±

26% of this evapotranspiration sourced from subsurface water (Good et al., 2015). The presence of shallow groundwater acts as a buffer, alleviating water stress in plants by enhancing transpiration rates and supplying water (Condon et al., 2020). In addition to plant transpiration, soil moisture dynamics, runoff generation, and atmospheric boundary layer development are all controlled by the groundwater table depth (Maxwell et al., 2016; Condon et al., 2013). Subsurface heterogeneity and topography are key determinants of groundwater distribution, with terrain-induced effects on vegetation being most

pronounced under limiting conditions of water, energy, or oxygen availability (Dai et al., 2003; Niu et al., 2005; Fang et al., 2022).

Despite these known dependencies, the hydrological processes in most dynamic global vegetation models (DGVMs), such as LPJ-GUESS, and land surface models (LSMs) within Earth System Models (ESMs), are generally depicted as one-dimensional (1-D, vertical) infiltration and evapotranspiration from the surface soil (e.g., Martín Belda et al., 2022). This modelling

approach oversimplifies groundwater connectivity, neglects lateral water redistribution and topographic gradients, and fails to capture the three-dimensional (3-D) surface-subsurface water dynamics that are essential to ecohydrological feedbacks (Oleson et al., 2018; Tang et al., 2014). It limits the accuracy of water partitioning between soil evaporation and plant transpiration, ultimately impairing hydrological and ecological predictive capability (Fan et al., 2019). Recent studies underscore the importance of addressing these limitations. For example, Lapides et al. (2024) emphasized the importance of incorporating the

bedrock vadose zone into LPJ-GUESS for accurate simulation of vegetation structure and function. Furthermore, an increasing body of research underscores that integrating dynamic groundwater processes is essential for closing the water and energy



balance in LSMs and ESMs (Pokhrel et al., 2012; Gleeson et al., 2012; Yang et al., 2020). Consequently, the integration of groundwater modules into ecosystem dynamics models has become a pressing need for accurately simulating the vegetation-climate interactions (Wang et al., 2024).

To bridge this gap, we developed an integrated modelling framework (PF-LPJG) coupling ParFlow (a fully distributed, physically based 3-D surface-subsurface hydrological model) with LPJ-GUESS (a dynamic global vegetation ecosystem model). This novel coupling explicitly captures the effects of lateral groundwater flow and topography on ecohydrological processes through comprehensive representation of feedbacks among groundwater dynamics, soil moisture connectivity, and vegetation productivity. We conducted simulation experiments in Europe's Danube River Basin using two model

configurations: (1) stand-alone LPJ-GUESS and (2) the fully coupled PF-LPJG system. Model outputs were rigorously evaluated against in situ observations and remote sensing data, enabling comprehensive assessment of hydrological and ecological performance metrics, highlighting the advantages of the coupled model in simulating water balance components. This integrated framework addresses critical limitations in current Earth system models and provides new mechanistic understanding of vegetation-groundwater responses to climate change.

The rest of the paper is organized as follows: Section 2 details the experimental design and data sources and the coupling framework between LPJ-GUESS and ParFlow. Section 3 presents validation results and model performance assessments in the study basins and discusses key mechanisms revealed by the coupled model, including interactions among groundwater, topography, and vegetation. Section 4 concludes the study and outlines future directions.

## 2 Method

### 2.1 LPJ-GUESS

The LPJ-GUESS model is a process-based dynamic global vegetation model (Smith et al., 2001, 2014). It integrates tree demographic processes and models competition for light, space, and soil resources among coexisting Plant Functional Types (PFTs) within a modelled area (patch) (Sitch et al., 2003, Tang et al., 2023; Chen et al., 2024). By simulating the establishment, growth, and mortality of age-based tree cohorts across multiple replicate patches, the model enhances its capability to represent

competitive mechanisms, population dynamics, community structure, carbon assimilation processes, and ecological succession across a heterogeneous landscape (Sitch et al., 2003; Smith et al., 2014).

The model includes a 1.5 m-deep active soil column (crucial for vegetation and biological processes), divided into fixed-depth layers of 10 cm each. Up to five snow layers can cover the soil surface, with a maximum snow depth of 1m. Below the soil layer, there are five padding layers with a total depth of 48 m, which are thermally active but hydrologically inactive. Soil

temperature is calculated daily by solving the heat transport equation in the soil column numerically, as detailed in (Tang et al., 2023; Gustafson et al., 2021).

In the LPJ-GUESS default hydrology model, each patch has an independent soil column with no lateral hydrological exchange between individual patches or grid cells and no representation of sub-grid terrain features. Precipitation and snowmelt are the



sources of water input at the top of this soil column, replenishing the upper soil layers (Gerten et al., 2004). Transpiration is
influenced by water availability in all soil layers, leaf and atmospheric demand, and the root distribution of existing cohorts
(Tang et al., 2018).

**2.2 ParFlow**

ParFlow is a fully integrated, physically based hydrological model that simulates three-dimensional surface and subsurface
water flow using high-performance parallel computing (Ashby and Falgout, 1996; Jones and Woodward, 2000; Kollet and
Maxwell, 2006). It was designed to simulate groundwater flow with overland flow in complex heterogeneous environments
and has been rigorously tested and validated (Maxwell et al., 2014). Utilizing a terrain-following grid system, ParFlow solves
the three-dimensional Richards' equation (Richards, 1931) to simulate variably-saturated flow in both confined and unconfined
aquifers, reaching depths of up to 1 km (Maxwell et al., 2013). This framework enables detailed modelling of near-surface
hydrodynamics as well as deeper subsurface processes.
The key feature of ParFlow is its seamless integration of subsurface and overland flow. This is achieved through a free-surface
boundary condition that integrates groundwater flow with surface runoff, capturing dynamic hydrological connectivity (Kollet
and Maxwell, 2006). Surface water bodies, such as streams and ponds, form within the simulation as a result of groundwater
convergence, saturation excess, or infiltration excess, rather than being imposed as predefined boundaries.
The numerical implementation of surface flow employs a standard upwind scheme for spatial discretization and utilizes an
implicit backward Euler differencing scheme (Jones and Woodward, 2001). Nonlinearities in the system are addressed using
a Newton-Krylov solution framework, where the Jacobian matrix is solved iteratively using Krylov subspace methods (Osei-
Kuffuor et al., 2014). This advanced numerical approach enhances the model's capability to simulate feedback between
groundwater, surface water, and the vadose zone with high accuracy. It also facilitates the incorporation of detailed soil
hydraulic properties and subsurface heterogeneity, enabling comprehensive simulations across spatial and temporal scales.
ParFlow solves Richards' equation for 3D variably saturated flow in the subsurface:

$$S_s S_W \frac{\partial \phi_p}{\partial_t} + \varphi \frac{\partial S_w}{\partial_t} = \nabla \cdot q + q_e(x) \tag{1}$$

$$q = -k(x)k_r(\psi_p)\nabla(\psi_p - z) \tag{2}$$

Where $\phi$ is the porosity [-], $\psi_p$ is the subsurface pressure head [L], $S_w$ is the water saturation [-], $S_s$ is the specific storage

coefficient [L$^{-1}$], $k$ is the saturated hydraulic conductivity [LT$^{-1}$], $k_r$ is the relative hydraulic conductivity [-], $q_s$ is the general

source/sink term [T$^{-1}$], $z$ is the depth below the surface [L], $q$ is the groundwater flux [L$^2$/T].

The boundary conditions are of the Neumann type:

$$-k(x)k_r\nabla(\psi_p - z) = q_{bc} \tag{3}$$



The fully integration of surface water and subsurface water is achieved through the free surface overland flow boundary condition: (1) The integration of surface water and groundwater is achieved through boundary conditions for free surface slope flow: $\psi_p = \psi_s = \psi$ .(2) Ensure the continuity of pressure at the surface and groundwater boundaries.

$$q_e(x) = \frac{\partial \|\psi,0\|}{\partial_t} - \nabla \vec{v} \|\psi,0\| - q_r(x) \tag{4}$$

$$-k(x)k_r\nabla(\psi - z) = \frac{\partial \|\psi,0\|}{\partial_t} - \nabla \vec{v} \|\psi,0\| - q_r(x) \tag{5}$$

Where $\psi_s$ is surface ponding depth [L], $\vec{v}$ is the depth averaged velocity [L/T].

The overland flow equations may be implemented into the Richards equation at the top boundary cell under saturated conditions. Using conditions of continuity of pressure ($\psi_p = \psi_s = \psi$) and flux ($q_e = q_{bc}$) at the ground surface, the $q_e(x)$ in Eq. (1) can be solved for Eq. (4). Equation (5) represents the surface water equation as a boundary condition for the Richards equation, assuming pressure continuity between the surface and subsurface regions. This indicates equal pressure at the surface, allowing the model to represent both regions simultaneously through Eq. (6). The head-dependent boundary condition accounts for the movement of the free surface of ponded water at the ground surface, enhancing the accuracy of the hydrological model by dynamically linking surface and subsurface flows.

$$S_s S_w \frac{\partial \psi_p}{\partial_t} + \varphi \frac{\partial S_w}{\partial_t} = \nabla \cdot [k(x)k_r(\psi)\nabla(\psi - z)] + q_s \tag{6}$$

**2.3 Coupling Model approach**

Groundwater-surface water interactions are critical in the hydrological process (Maxwell et al., 2016). As mentioned in Sect. 2.1, LPJ-GUESS simplifies these interactions and overlooks the heterogeneity of topographical conditions (three-dimensional flow movement). To address this, we coupled the three-dimensional surface-subsurface hydrological model ParFlow with the process-based vegetation dynamics and ecosystem model LPJ-GUESS, enabling comprehensive simulation of the hydrological cycle from bedrock through plant to the atmosphere. Figure 1 illustrates the conceptual design of the coupled PF-LPJG framework and the integration strategy that allows for dynamic interactions between the hydrological and ecological components.

Several modifications on the LPJ-GUESS structure were made before coupling: (1) the original fixed-depth soil discretization (15 layers at 10 mm each) was replaced with four variable-depth layers (0.1, 0.3, 0.6, and 1 m), totaling 2 m, to align with the discretized soil columns used by ParFlow; (2) layer-specific soil property inputs (sand, clay, and silt content) were introduced to represent vertical heterogeneity in hydraulic properties; (3) Soil water content simulated by ParFlow was used to overwrite the internal soil moisture states in LPJ-GUESS; (4) an MPI-based coupler was embedded within the main LPJ-GUESS framework (framework.cpp), enabling synchronous, two-way communication between the two models while preserving their native solvers.

The coupling methodology involved replacing the one-dimensional soil moisture and runoff module in LPJ-GUESS with the three-dimensional surface-subsurface flow module in ParFlow. Daily, ParFlow provides LPJ-GUESS with updated soil



moisture and surface/subsurface runoff fields. At the same time, LPJ-GUESS returns the net atmospheric water input

(precipitation minus evapotranspiration, P-ET) as an upper boundary condition to ParFlow. This two-way exchange ensures a

fully interactive and temporally consistent coupling of hydrological and ecological processes.

The interaction occurs in the top soil layers (0-2m). During the coupled simulation, LPJ-GUESS simulates one patch of a

single stand type with the probabilistic patch-destroying disturbance turned off to prevent the complete loss of vegetation from

an entire grid cell in a single event. Each patch in LPJ-GUESS aligns with the uppermost layer of a vertical column in ParFlow,

ensuring consistency between the soil columns in both models.

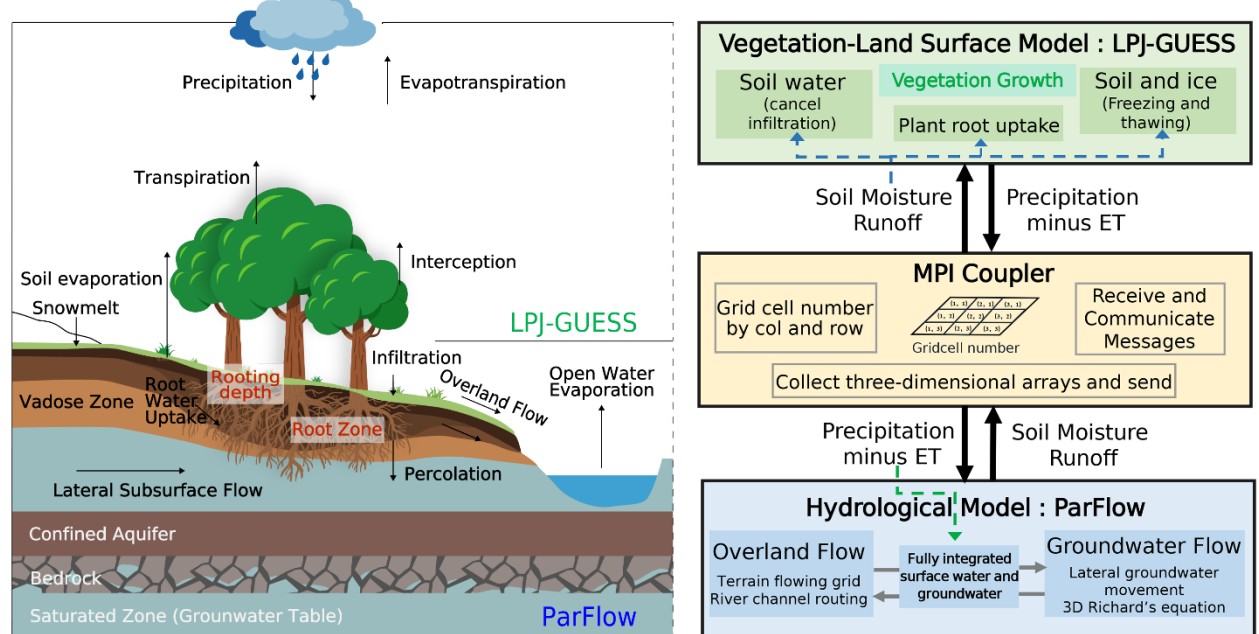

**Figure 1.** Schematic representation of the coupled vegetation dynamics-surface-subsurface flow system. Blue dash arrows indicate outline fluxes calculated and passed by ParFlow, green dash arrows represent net water input (P-ET) computed and passed by LPJ-GUESS.

### 2.4 Data sets for model parameterizations and evaluations

The Danube River Basin encompasses both humid and semi-arid regions, with diverse topography from mountains to plains,

making it ideal for studying hydrological processes. The basin is also relatively unaffected by intensive human activities, like

groundwater pumping and large-scale irrigation, and it has been extensively studied in a lot of research (Naz et al., 2023; Zhou

et al., 2024).

### 2.4.1 Meteorological data

For the Danube River Basin, the ERA5-LAND dataset with a horizontal spatial resolution of 0.1°× 0.1° from 1980 to 2018

served    as    the    meteorological    forcing    data    and    is    available    from    the    Copernicus    Climate    Data    Store

(https://doi.org/10.24381/cds.e2161bac, Muñoz Sabater, 2019). The meteorological data included precipitation, 10 m u-



component of wind, relative humidity, surface downwelling shortwave radiation, surface air pressure, and 2 m temperature, all aggregated to a daily time scale using the CDS daily aggregation method.

### 2.4.2 Soil data

The coupled model's top 2 meters are divided into four soil layers (0.1, 0.3, 0.6, and 1.0 m, from top to bottom). The fractions of sand, clay, silt, organic carbon, pH, and bulk density in each layer were derived from the WISE database, which provides data representative of the soil units by the FAO Soil map and the Harmonized World Soil Database (HWSD) (Batjes 2016). The spatial resolution of the soil grid is 30" × 30", resampled to 10 km using the dominant soil type method. These data were aggregated to match the model layers and served as the layer-specific soil data input for both ParFlow and LPJ-GUESS to ensure consistency in physical soil representation across the coupled modelling framework.

### 2.4.3 Subsurface data

For each hydrogeological unit in the study area, an indicator file was required for ParFlow, including hydraulic conductivity, van Genuchten parameters, and porosity. Hydrolithologic categories in the indicator file were reclassified from the permeabilities of GLHYMPS 1.0 dataset (Gleeson et al., 2014), and the bedrock classification was revised using the hydrogeological maps of the Danube River Basin (Duscher et al., 2015).

To better represent actual bedrock conditions, a variable-depth flow barrier was set between each bedrock aquifer, with the reference bedrock depth values from Shangguan et al. (2017). The flow barrier reduces vertical water flux across grid interfaces by a factor of 0.001, enabling the model to distinguish between shallow unconfined aquifers and deep confined aquifers (Tijerina-Kreuzer, 2024). Additionally, an e-folding method was applied to the subsurface, which reduces hydraulic conductivity with depth and slope using a depth-dependent factor (Fan et al., 2007).

### 2.4.4 Land Cover and Manning's n

Land cover classification was derived from the MODIS Land Cover Type product (MCD12Q1, Version 6.1), which provides global coverage at a 500 m spatial resolution (Friedl and Sulla-Menashe, 2022). This classification was derived from the Land Cover Type: The International Geosphere-Biosphere Programme (IGBP) global classification scheme and was resampled to 10 km resolution using the majority method (Foster et al., 2019) to match the PF-LPJG model resolution.

To parameterize spatially variable surface roughness, Manning's n coefficients were assigned based on land cover types, with further adjustments in stream order networks. River channels were delineated using the PriorityFlow algorithm with a threshold drainage area of 50 km², and stream orders were assigned following the Strahler classification system (Strahler, 1957). Manning's coefficients for streamflow were assigned to decrease with increasing stream order, reflecting reduced channel resistance in higher-order rivers (Gochis et al., 2015).



### 2.4.4 Topography and rivers

The Digital Elevation Model (DEM) data for the study area were obtained from the MERIT Hydro IHU dataset. MERIT Hydro IHU is a global hydrography dataset upscaled from the 3" resolution MERIT Hydro using the Iterative Hydrography Upscaling

(IHU) algorithm (Eilander et al., 2021). MERIT Hydro IHU provides multiscale topographic information at spatial resolutions of 30" (~1 km), 5' (~10 km), and 15' (~30 km). For this study, the 5' DEM was selected and resampled to 10 km using the minimum resampling method in QGIS (Zhang et al., 2021).

To ensure a fully connected drainage network in the study area, the upscaled DEM was preprocessed with the priority flow algorithm to specify the slope in the x and y directions (Condon and Maxwell, 2019). This algorithm enforces drainage in the

primary east, west, south, and north directions (D4 directions) because the model's hydrological simulations are based on partial differential equations (PDEs) (Zhang et al., 2021). The slope was calibrated based on the drainage area in MERIT Hydro IHU for the Danube River basin and further refined by the parking lot tests to accurately determine river network locations and eliminate anomalous ponding points (Bhaskar, 2010).

### 2.4.5 Datasets for evaluation

The modelled daily streamflow was compared to river flow observations from the Danube River Basin, obtained from the Global Runoff Data Center (GRDC, obtained at https://grdc.bafg.de/, last access: 21 August 2025). The simulated surface SM at 0.1m depth was evaluated by comparison with the global satellite-based estimates from the European Space Agency Climate Change Initiative Soil Moisture product, ensuring that NaN values in the original data were excluded from the correlation calculation (ESA CCI-SM; Dorigo et al., 2017). To match the model-simulated SM with the dataset, we interpolated the ESA

CCI-SM data from 0.25° to 0.1° (~10 km) resolution using bilinear interpolation. The validation of ET data was performed using Global Land Evaporation Amsterdam Model (GLEAM) 4.2 datasets, which employ the Priestley-Taylor equation and other algorithms to estimate ET separately for both soil and vegetation (Martens et al., 2017).

The WTD validation data include two datasets: (1) a global gridded WTD benchmark produced by Fan et al. (2013), which couples a Darcy-based 2D lateral flow scheme with a 1D vertical unsaturated flow process described by the Richards equation,

and (2) in situ groundwater observations from monitoring wells across the Danube Basin (Fan et al., 2013).

### 2.5 The modelling spin-up

The PF-LPJG coupled model was simulated over the Danube River Basin with a horizontal resolution of 10 km, resulting in a grid of 161 columns × 82 rows. The subsurface was discretized into 10 vertical layers of increasing thickness from top to bottom: 0.1, 0.3, 0.6, 1, 5, 10, 10, 10, 25, and 50 m, capturing the transition from shallow soil to deep aquifer systems. No-

flow boundary conditions were prescribed along the lateral and bottom boundaries, while the upper boundary employed a kinematic wave formulation to simulate overland flow processes.



To ensure a physically consistent initial condition, a two-phase spin-up was adopted for the coupled system. First, ParFlow and LPJ-GUESS were spun up independently. For ParFlow, the model was initialized using the long-term (1980-2018) average net water input (P-ET), with the ET derived from multi-year average LPJ-GUESS simulations over the Danube River Basin.

The initial groundwater table was set at a depth of 5 m, removing all positive pressure heads and prohibiting overland flow. The model runs until the change in groundwater storage stabilizes less than 1 % of the potential recharge(P-ET), allowing the subsurface system to reach equilibrium over the spin-up period (Seck et al., 2015; Ajami et al., 2014). In the second phase, overland flow was activated, and ParFlow was run for a prolonged duration to allow the coupled surface-subsurface flow system to reach hydrological steady state (the total storage change was less than 3 % of the P-ET) and generate a stable stream

network structure. For LPJ-GUESS, a 500-year spin-up process using repeated 30-year meteorological forcing (1980-2010) was used to estimate vegetation composition and biomass at dynamic equilibrium in response to climatic and edaphic conditions. Soil organic matter is additionally spun-up offline at year 300 of the standard LPJ-GUESS spin-up for 40,000 years using repeated 100-year soil moisture and temperature status and litter input from years 200-300.

Following the individual spin-ups with saved state variables, the fully coupled PF-LPJG model was initialized and driven by

repeating the meteorological data of 1980 for 10 consecutive years to reach dynamic equilibrium of the two models before starting the formal simulation (1980-2018). The spin-up period precedes significant human activities such as dam construction, hydroelectric power plants, and groundwater pumping in the modelling domain.

## 2.6 Model evaluation

To systematically evaluate the performance of the coupled model PF-LPJG in streamflow simulation, we use four statistical

metrics between observed and simulated values: Spearman rank correlation coefficient (to assess the model's ability to capture streamflow trends), Kling-Gupta Efficiency (KGE, to evaluate holistic model behaviour), Percent Bias (PBIAS, to reflect systematic bias), and Root Mean Square Error-observations Standard deviation Ratio (RSR, to assess relative error). These metrics were selected to provide a comprehensive assessment of the model's performance against in situ, remote sensing observations, and reanalysis datasets, assessing both the accuracy of the simulated values and their temporal and spatial

dynamics.

Spearman's $\rho$ is a non-parametric measure of monotonic association between two variables.

$$\rho = 1 - \frac{6\sum_{i=1}^{n} d_i^2}{n(n^2-1)} \tag{7}$$

Where $d_i$ is the difference between the ranks of observed and simulated values for the observation, $n$ is the total number of values.

*KGE* explicitly decomposes model performance into correlation, bias, and variability components to evaluate the strengths and weaknesses of the model.

$$KGE = 1 - \sqrt{(r-1)^2 + (\beta-1)^2 + (\gamma-1)^2} \tag{8}$$

Where $r$ is the Pearson correlation coefficient between observed and simulated values, $\beta$ and $\gamma$ are the bias ratio and variability ratio.



*PBIAS* measures the average tendency of the simulated values to be larger or smaller than their observed counterparts.

$$PBIAS = 100 \times \frac{\sum_{i=1}^{n}(s_i - o_i)}{\sum_{i=1}^{n} o_i} \tag{9}$$

Where $s_i$ is simulated value at time $i$, $o_i$ is observed value at time $i$.

*RSR* is the ratio of the root mean square error (RMSE) to the standard deviation of the observed data.

$$RSR = \frac{\sqrt{\frac{1}{n}\sum_{i=1}^{n}(s_i - o_i)^2}}{\sigma_o} \tag{10}$$

Where $\sigma_o$ is the standard deviation of observed values, $i$ is the total number of observations.

## 3 Results and Discussions

### 3.1 Evaluation of streamflow

This section assesses the model's ability to reproduce observed hydrological behaviour across seven GRDC stations representative of diverse catchment sizes (large, medium, small) and climatic conditions in the Danube basin. These stations
were selected based on their spatial representativeness in terms of geographic distribution and catchment characteristics, and the availability of complete daily discharge data for the period 1980-2018. The PF-LPJG coupled model, by explicitly coupling LPJ-GUESS with a physically based groundwater module via ParFlow, offers substantial improvements in capturing streamflow dynamics compared to its uncoupled counterpart. As shown in the boxplots of performance metrics for the seven basins, the PF-LPJG model systematically outperforms the stand-alone LPJ-GUESS model across all key metrics, with higher
trend consistency and correlation, and lower bias and relative error (Fig. 2). Notably, over 80% of the gauging stations report KGE values exceeding 0.5 and Spearman coefficients above 0.8, demonstrating the coupled model reliably captures the temporal consistency and responds to hydrological variability (Fig. 3). The improvements seen in PF-LPJG are particularly salient at major downstream locations (e.g., CEATAL IZMAIL), where all performance metrics demonstrate clear enhancement. This advancement reflects the well-documented limitation in the current dynamic vegetation models (DGVMs),
which typically exhibit poor streamflow simulation performance when groundwater dynamics and lateral flow processes are inadequately represented (Yang et al., 2015; Hong et al., 2025). By integrating subsurface hydrological processes, PF-LPJG addresses a long-standing limitation in ecohydrological modelling (Guillaumot et al., 2022).

Despite these improvements, both PF-LPJG and LPJ-GUESS exhibit limitations under hydroclimatic extremes (wet year: P-ET above the long-term mean; dry year: P-ET below the long-term mean). During anomalously wet years (e.g., 2010, 2014),
both PF-LPJG and LPJ-GUESS overestimate peak flows and underestimate low flows during dry years (e.g., 2003, 2011). Nevertheless, PF-LPJG exhibits a substantially narrower bias range (PBIAS: 0.2-0.4) and improves relative error metrics (RSR: 0.9-1.3), compared to LPJ-GUESS (PBIAS: 0.3-0.5; RSR: 1.7-2.0) (Fig. 2), suggesting that coupled groundwater dynamics contribute to more stable and physically consistent responses.



The PF-LPJG model exhibits strong seasonality in performance, with higher consistency during spring snowmelt and summer
rainfall-driven peaks. In dry years (1983, 1987, 2003, 2006, and 2015), more than four gauging stations, including upstream
and downstream locations (CEATAL IZMAIL, ORSOVA, SZEGED, KIENSTOCK), show both R² and Spearman's ρ > 0.9,
and KGE > 0.5. This consistency indicates improved simulation of seasonal runoff processes across the basin.    Conversely,
the model's skill deteriorates in the year 2001, with the weakest model performance across the basin (R² and Spearman's ρ are:
0.1-0.3; KGE ≈ 0 at multiple stations).  Although the year 2001 did not rank among the extremes of precipitation, temperature,
or ET, its poor performance may reflect lagged hydrological effects from 2000, a dry year with the third lowest P-ET value.
This likely reflects disrupted hydrological connectivity between soil moisture and active surface water components (Good et
al., 2015), underscoring the need for improved parameterizations and potential assimilation of satellite-derived soil moisture
and groundwater storage data (Clark et al., 2015).

From a spatial perspective, PF-LPJG performs better in medium-to-large catchments (>10,000 km²), where hydrological
signals are less sensitive to local-scale heterogeneity. Conversely, smaller basins (e.g., LUNGOCI) exhibit poor simulation
performance (KGE = 0.046; PBIAS = 49.1%) (Fig. 3), likely due to coarse spatial resolution (10 km), unresolved
anthropogenic influences (e.g., reservoirs, pumping), and scaling issues inherent in DGVMs (Wood et al., 2011). Building
high-resolution routing schemes or regional hydrological models could mitigate such deficiencies.

The PF-LPJG model's ability to mitigate summer low-flow underestimation during dry years and to better capture peak timing
in wet years demonstrates the role of groundwater buffering. These improvements reflect that groundwater contributions can
help delay soil moisture depletion, maintain streamflow, and support plant growth demand under drought conditions.
Especially, subsurface water stores wet-season infiltration and routes it to streams during dry periods, resulting in a fairly
constant baseflow year-round and sustaining summer discharge a process poorly represented in conventional DGVMs
(Berghuijs et al., 2024; Condon et al., 2019).

These improvements stem from the integrated, physically based groundwater-surface water interactions enabled by ParFlow.
Unlike empirical routing schemes, the coupling model resolves the expansion and contraction of surface water channels purely
from flux balances and terrain properties, without requiring any a priori specification of inundated areas (i.e., rivers, lakes, and
wetlands). This is consistent with other advanced models such as CLM-ParFlow and mHM-HGS, which have shown similar
strengths in capturing low-flow dynamics and recession curve representation (Maxwell et al., 2005; Kumar et al., 2013).

Nonetheless, persistent peak-flow overestimations in some regions remain, largely due to the limitations of coarse spatial
resolution. The 10 km resolution grid leads to a smoothing of topographic gradients, which flattens slopes, shortens flow
pathways, and raises groundwater heads, ultimately resulting in unrealistically rapid and voluminous runoff responses.
Furthermore, the Danube River basin's cold-climate characteristics, such as suppressed evapotranspiration and intense spring
snowmelt, contribute to seasonal runoff amplification (Zhou et al., 2024). This hydrological signature further exacerbates peak
overestimation, particularly when land surface energy partitioning is poorly resolved.





In summary, the PF-LPJG model represents a notable advancement in large-scale land surface hydrology by embedding groundwater-surface water interactions into a DGVM. This integration allows for improved representation of both mean flow conditions and hydrological extremes, even without empirical calibration or high-resolution.

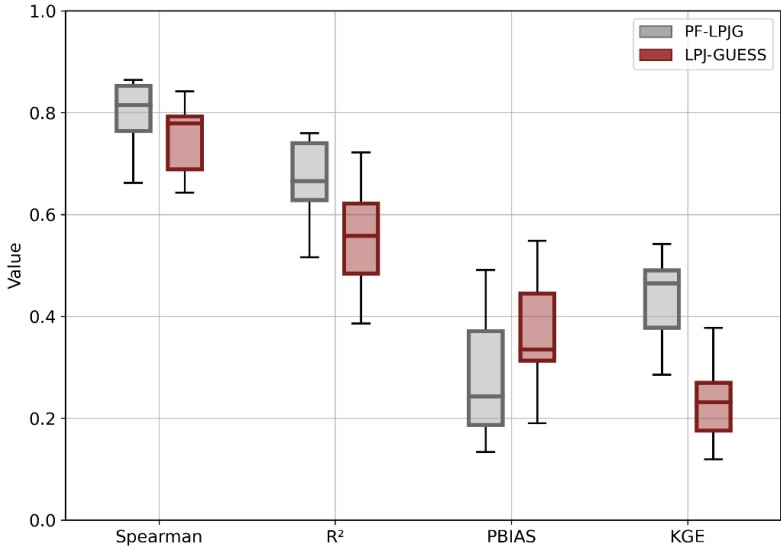

**Figure 2.** Boxplots of performance metrics (Spearman coefficient, R², PBIAS, RSR, and KGE) for daily streamflow simulated by PF-LPJG and LPJ-GUESS against GRDC observations.



**Figure 3.** Comparison of monthly discharge simulated by the coupled model PF-LPJG and the stand-alone LPJ-GUESS at seven gauging stations from the GRDC dataset.





## 3.2 Evaluation of Evapotranspiration

Figure 4a-f presents the spatial distribution of mean annual ET for the period 1980-2018 from PF-LPJG, LPJ-GUESS, and GLEAM, along with their inter-model differences. PF-LPJG successfully reproduces the similar spatial patterns of ET across the basin, capturing gradients from the drier western and northeastern regions to the wetter central zones. However, the model tends to underestimate ET in colder northern and southwestern regions (where the mean annual temperature is < 0°C) and overestimate ET in the humid central lowlands. These biases are consistent with prior studies using ParFlow in Europe (Naz et al., 2023), where ET discrepancies in river channels were primarily due to ParFlow's default assumption that soil saturation is set to 1.0 in river channel cells.

We further analysed monthly ET outputs from PF-LPJG and LPJ-GUESS. Across most of the Danube basin, differences between the two models are small (mean bias: -0.007 mm day$^{-1}$). The PF-LPJG model shows a slight improvement in agreement with GLEAM data when river-channel grid cells are excluded from the analysis. Specifically, the mean difference between PF-LPJG and GLEAM (0.101 mm day$^{-1}$) is marginally smaller than that of LPJ-GUESS (0.108 mm day$^{-1}$), suggesting a modest enhancement in ET realism following hydrological coupling (Fig. 4g).

The model's performance in simulating seasonal ET dynamics was evaluated using cumulative distribution functions (CDFs) (Fig. 4h). The seasonal discrepancies between simulated and observed ET vary notably throughout the year. During autumn (SON) and winter (DJF), mean ET deviations are relatively minor (0.045 and -0.163 mm day$^{-1}$, respectively), indicating reasonable agreement with GLEAM data. However, during spring (MAM) and summer (JJA), larger positive deviations emerge, with mean differences of 0.226 and 0.464 mm day$^{-1}$ for xx and yy, respectively. These summer overestimations are most prominent along the riparian zones of the river network, which may be attributed to increased soil moisture availability and elevated temperatures, both of which promote enhanced ET (Fig. 4h).

Importantly, the PF-LPJG model appears particularly sensitive to soil texture and hydraulic resistance under dry conditions. In arid regions of the western and northeastern basin, ET rates are lower and closely aligned with GLEAM estimates, indicating that water availability limits evapotranspiration. In contrast, certain flat areas with coarse soils exhibit enhanced ET in PF-LPJG, likely due to increased infiltration capacity and accumulation of shallow soil moisture, which amplifies bare-soil evaporation (Miguez-Macho et al., 2012). This effect is exacerbated by the model's spatial resolution (~10 km), which smooths topographic gradients and can lead to artificial surface saturation and restricted drainage in low-slope areas (Jefferson and Maxwell, 2015).

Further analysis highlights the model's sensitivity to soil texture and hydraulic conductivity, particularly under dry conditions. In arid areas of the western and northeastern basin, ET in PF-LPJG is constrained by water availability and closely matches GLEAM estimates. Conversely, in flat regions with coarse soils, the model tends to overestimate ET due to limited infiltration capacity and resulting shallow soil moisture accumulation, which enhances bare-soil evaporation (Fan et al., 2019). This effect is exacerbated by the coupled model's spatial resolution (~10 km), which smooths topographic variation, leading to artificial saturation zones and restricted drainage in low-relief areas (Jefferson and Maxwell, 2015).



Although some positive ET biases persist near river channels, driven by the tendency of ParFlow to saturate riparian zones, PF-LPJG on the whole demonstrates improved spatial and seasonal representation of ET compared to the stand-alone LPJ-GUESS model (Fig. 4e, g, h). These improvements are evident across a range of hydroclimatic settings within the Danube basin and highlight the benefits of integrating physically based hydrological processes into terrestrial ecohydrological models. The results underscore the potential of PF-LPJG as a tool for integrated assessments of land-atmosphere feedbacks and water use dynamics, especially under scenarios of increasing climate variability.

**Figure 4.** Spatial distribution of mean annual ET. **(a-c)** and inter-model differences. **(d-f)** for the period 1980-2018 as simulated by the coupled model PF-LPJG, the stand-alone model LPJ-GUESS, and the GLEAM 4.2 datasets across the Danube River Basin. **(g)** Histogram of mean annual ET from GLEAM v4.2, LPJ-GUESS, and PF-LPJG. **(h)** Cumulative distribution function (CDF) comparison of seasonal (DJF, MAM, JJA, SON) ET comparing PF-LPJG and GLEAM v4.2a.



### 3.3 Evaluation of Soil Moisture

To evaluate the performance of the PF-LPJG model in simulating soil moisture dynamics, we compared the simulated volumetric soil water content at 0.1 m depth with satellite-based estimates from the ESA Climate Change Initiative Soil Moisture product (CCI-SM; Dorigo et al., 2017).

Figure 5 illustrates the spatial pattern of surface volumetric soil moisture simulated by PF-LPJG and observed by CCI-SM. In humid regions of the Danube Basin, PF-LPJG tends to overestimate near-surface soil moisture, especially during wet months

(Fig. 5c). This overestimation may be attributed to ParFlow's simulation of lateral and vertical redistribution processes, which enhances surface saturation, particularly in low-relief areas or near river networks (Maxwell and Condon, 2016). The PF-LPJG model captured spatial heterogeneity effectively, owing to its ability to simulate deeper infiltration and slower recharge processes that dominate soil moisture dynamics in water-limited settings. These features enabled PF-LPJG to better simulate deeper soil moisture in arid regions, as indicated by higher spatial correlation with CCI-SM (Fig. 5d).

Monthly anomalies of surface soil moisture aggregated across the Danube Basin over the period 1980-2018 are shown in Fig. 6b. The PF-LPJG exhibits good temporal consistency with CCI-SM anomalies (Spearman's ρ = 0.51, RMSE = 0.73 cm³/cm³), indicating an improved capacity to capture both seasonal and interannual variability. The Violin plots (Fig. 6a) comparing monthly soil moisture distributions between PF-LPJG and CCI-SM show that PF-LPJG generally overestimates soil moisture, particularly in wet months. Anomalously low soil moisture anomalies in early 1985 simulated by PF-LPJG were likely caused

by extremely low temperatures in January and February of that year, with mean basin-wide temperatures below 0°C. These conditions resulted in widespread snow accumulation, reduced meltwater availability, and suppressed evapotranspiration. These cold-season hydrological effects are consistent with previous findings on snow-dominated basins (Ionita et al., 2018). All of which were reflected in the PF-LPJG simulation.

Model performance under hydroclimatic extremes was further evaluated by analysing cumulative distribution functions (CDFs)

of RMSE and Spearman correlation coefficient values for dry and wet months (Fig. 6c, d). The results indicate that PF-LPJG significantly improved soil moisture simulation during dry periods, with higher correlation and lower error compared to LPJ-GUESS. This enhancement is attributed to the representation of soil-groundwater feedbacks, including infiltration and subsurface storage buffering, which are largely absent in traditional DGVMs (Fan et al., 2019). During drought events, PF-LPJG exhibits stronger vertical flux dynamics and a clearer depletion signal in surface soil moisture, more closely aligned with

CCI-SM anomalies. Additionally, the dynamic interaction between vegetation available water and root-zone uptake water reinforces the model's responsiveness to water stress, as vegetation dynamics directly affect soil moisture redistribution and evapotranspiration efficiency (Bonan et al., 2019; Fan et al., 2017). However, its advantage is less evident under wet conditions. This may be attributed to the reduced role of subsurface processes when soils are near saturation, limiting the contribution of explicit groundwater representation.

At the spatial scale, PF-LPJG effectively captures heterogeneity in soil moisture, benefiting from ParFlow's three-dimensional representation of groundwater flow and topographic controls (Kuffour et al., 2020). In the northwestern basin, where annual



precipitation exceeds evapotranspiration, elevated groundwater tables result in persistent surface saturation and slight overestimations of soil moisture. Nevertheless, high spatial correlation with CCI-SM, particularly under water-limited conditions, indicates that PF-LPJG realistically simulates the dominant controls on soil water dynamics across diverse hydroclimatic gradients (Soltani et al., 2022).

As a whole, PF-LPJG captures both short-term soil moisture variability and the legacy effects of wet or dry conditions, providing strong seasonal and interannual memory in soil moisture.

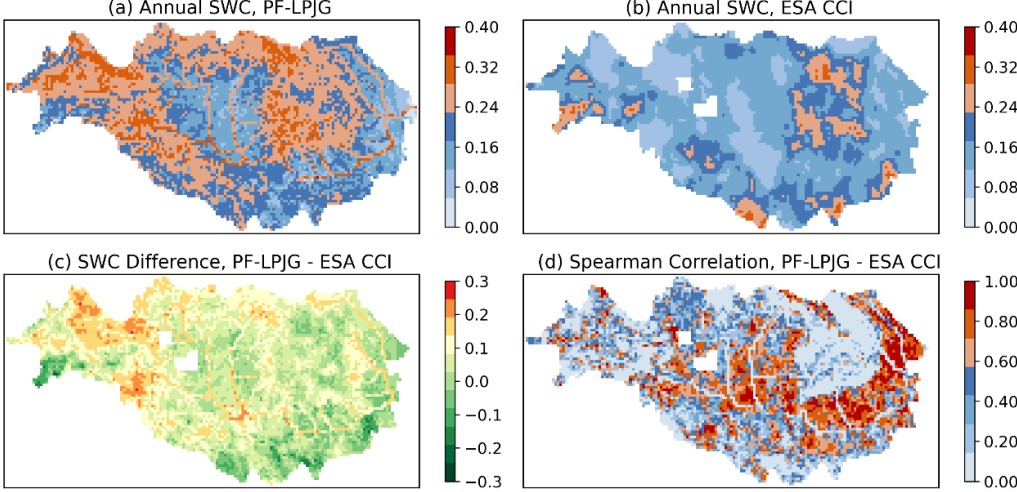

**Figure 5. (a-b)** Evaluation of time-averaged surface soil moisture (SM) simulated by the coupled model PF-LPJG against the ESA CCI-SM dataset during 1980-2018. **(c)** Mean difference and **(d)** Spearman's correlation in SM between PF-LPJG and ESA CCI-SM.





**Figure 6**. **(a)** Violin plots of the spatial distribution of time-averaged surface soil moisture (SM) from PF-LPJG and ESA CCI-SM over the Danube River Basin during 1980-2018. **(b)** Spatially aggregated monthly anomalies of surface SM from PF-LPJG and ESA CCI-SM. **(c-d)** Cumulative distribution functions (CDFs) of RMSE and Spearman's ρ for surface SM in wet (4-9) and dry months (1-3, 10-12), comparing PF-LPJG and LPJ-GUESS simulations against ESA CCI-SM.

## 3.4 Evaluation of Water Table Depth

WTD is a critical hydrological variable that governs plant-groundwater interactions, particularly under conditions of water limitation and hypoxia (Fan et al., 2017; O'Connor et al., 2019). It plays a dual role-acting as supplementary water source that sustains transpiration during dry periods, and influencing plant productivity and community composition across diverse climatic regimes. Beyond its ecohydrological relevance, WTD provides a stringent diagnostic for evaluating groundwater model performance due to its spatial and temporal sensitivity to hydrological forcing (Yang et al., 2023).



To evaluate the ability of the PF-LPJG model to simulate groundwater dynamics, we compared modelled WTD outputs against two independent datasets: (1) a global gridded WTD benchmark produced by Fan et al. (2013), and (2) in situ groundwater observations from monitoring wells across the Danube Basin. As shown in Fig. 8a, the PF-LPJG model achieved a residual-

440     based RSR (RMSE divided by the standard deviation of observations) of 1.31, which is substantially lower than the RSR of 4.46 from the Fan et al. simulation. The histogram of the absolute errors between PF-LPJG and observational data (Fig. 8b) indicates that for most well points (48 observation points), the difference between the simulated WTD value and the observed WTD value is within the 0-5 m range. The evaluation shows that the PF-LPJG model performs well in reproducing observed WTD dynamics.

445     In addition, the cumulative distribution functions (CDFs) of WTD from monitoring stations revealed that PF-LPJG better captured the range of water table fluctuations, particularly in regions with deeper water tables. Figure 8c illustrates the monthly mean time series of WTD simulated by PF-LPJG at selected monitoring sites. Seasonal groundwater dynamics were well captured, with water tables rising in response to spring recharge (typically peaking in May) and gradually declining through the dry summer months. At sites where WTD exhibited low interannual variability and smooth transitions, the model

450     effectively replicated this stability, suggesting that PF-LPJG can reproduce both dynamic and quasi-steady groundwater regimes (Fig. 8c).







**Figure 7. (a)** Cumulative distribution function (CDF) of WTD as simulated by PF-LPJG and Fan's model with observed WTD data. **(b)** Histogram plots of the absolute errors between PF-LPJG simulated minus observed WTD. **(c)** Monthly mean time series of simulated WTD by PF-LPJG at selected monitoring sites.



### 3.5 Improvement of Evapotranspiration Partitioning

To disentangle the hydrological controls on (ET), we decomposed ET into bare-soil evaporation (E) and plant transpiration (T). While E is predominantly determined by near-surface soil moisture availability, T reflects root-zone water uptake and is modulated by vegetation phenology and physiological traits (Maxwell and Condon, 2016; Li et al., 2019). To assess the role of groundwater in regulating the partitioning between E and T, we compared monthly cumulative values of T, E (Fig. 9a), and their ratio (T/E) from three data sources (Fig. 9b): the coupled PF-LPJG model, the standalone LPJ-GUESS model, and the GLEAM v4.2 dataset.

The results reveal that PF-LPJG captures the seasonal dynamics of T/E with higher fidelity to GLEAM, especially during summer months characterized by elevated atmospheric evaporative demand (Fig. 9). In contrast, LPJ-GUESS exhibits unrealistic T/E peaks and severely constrained E during these periods, indicative of a rapid depletion of shallow soil moisture in the absence of upward capillary fluxes from the groundwater table, limiting E and leading to unstable T/E ratios.

The incorporation of a physically based groundwater component in PF-LPJG enhances surface moisture supply which in turn improves plant water utilization during dry or hot periods, thereby supporting bare-soil evaporation and stabilizing the T/E ratio (Fig. 9b). Although T remains relatively constant across simulations, small increases in evaporation enabled by groundwater can significantly lower the T/E ratio, highlighting the groundwater contributions can noticeably shift the balance of ET partitioning. This behavior reflects the groundwater dependence of the ET partitioning process, as previously observed in empirical studies (Condon et al., 2020; O'Connor et al., 2019; Davison et al., 2018). The agreement with seasonal dynamics between PF-LPJG estimates and GLEAM reinforces the hypothesis that subsurface-plant hydrological connectivity plays a critical role in mediating the E-T partitioning, particularly under climatic extremes.

Nonetheless, limitations remain in the model representation. The use of a fixed rooting depth (0-2 m) and a static exponential root distribution (controlled by PFT specific parameter (Root_Beta)) in LPJ-GUESS does not account for root plasticity in response to vertical soil moisture gradients or fluctuating groundwater levels. Such simplifications constrain the model's ability to represent vegetation strategies under water stress, especially for deep-rooted species and phreatophytes (Feddes et al., 2001; Warren et al., 2015). Future work may incorporate dynamic root modelling to allow root distribution to respond to variations in soil water availability (Lu et al., 2019).







**Figure 8.** Monthly cumulative values of transpiration (T) and evaporation (E) **(a)**, and the ratio (T/E) **(b)** as simulated by the coupled model PF-LPJG, the stand-alone model LPJ-GUESS, and the GLEAM v4.2 dataset.

## 4 Conclusions

This study introduces the first basin-scale, long-term implementation of the fully coupled PF-LPJG modelling framework, which physically integrates the three-dimensional ParFlow hydrological model with the LPJ-GUESS dynamic vegetation model across the Danube Basin at 10 km spatial resolution. We rigorously evaluated the model using comprehensive statistical metrics (KGE, Spearman's ρ, PBIAS, RSR, R²), benchmarking performance against the stand-alone LPJ-GUESS model, and further evaluated against different observational datasets. The PF-LPJG framework demonstrated substantial improvements in runoff and soil moisture simulations compared to standalone LPJ-GUESS. The coupled model achieved strong performance (R > 0.8 and KGE > 0.5) without parameter calibration at four out of seven gauging stations, including both headwater and outlet sites, with particularly remarkable improvements at basin outlets. These enhancements highlight the role of the ParFlow



variably saturated flow solver in enhancing lateral subsurface redistribution and drainage processes-critical mechanisms for large-scale hydrological and ecosystem modelling. Moreover, PF-LPJG also improved the partitioning of evapotranspiration. The simulated transpiration-to-evaporation (T/E) ratios aligned more closely with observations from the GLEAM v4.2 dataset. These improvements reflect the model's capacity to sustain evapotranspiration under hydroclimatic stress through capillary rise and lateral groundwater convergence. Unlike LPJ-GUESS, which lacks representation of groundwater-mediated soil moisture buffering, PF-LPJG maintains surface moisture availability during dry periods, thereby enhancing the realism of drought response and vegetation-soil moisture coupling. These dynamics also contributed to enhanced simulation of soil moisture seasonal anomalies and spatial variability, particularly in low-relief and groundwater-connected areas. Although PF-LPJG tended to simulate shallower water table depths (WTDs) than observed, its spatial WTD patterns exhibited greater agreement with in-situ data and outperformed benchmark simulations from Fan et al. (2013).

This study presents the implementation of the fully coupled PF-LPJG model to simulate 38 years of hydrological and vegetation dynamics at 10 km resolution across the Danube River Basin, incorporating an explicit representation of lateral groundwater flow. While this marks a significant achievement, several inherent limitations in the current model configuration must be acknowledged. The primary uncertainties arise from coarse soil and hydrogeological parameterizations, smoothed topography that affects local hydrodynamics, and simplified root water uptake schemes within LPJ-GUESS, and LPJ-GUESS can only be run with a single patch per ParFlow grid cell. Future work should aim to incorporate dynamic root-water interactions to enhance the realism of model simulations. Overall, the coupled PF-LPJG model represents a step forward in physically-based ecohydrological modelling, enabling more rigorous assessments of groundwater-vegetation interactions at regional scales. The modelling framework also offers valuable potential for investigating vegetation resilience to enhanced drought conditions, wetland dynamics, and carbon dynamics under climate change.

**Code and data availability**

The open-source ParFlow version 13.0 is available at https://github.com/parflow/parflow/releases/tag/v3.13.0 (last access: 21 August 2025; Smith, 2025). The LPJ-GUESS version used in this study is archived on Zenodo at (https://doi.org/10.5281/zenodo.8070582, Lindeskog et al., 2017). The coupled model ParFlow-LPJ-GUESS (PF-LPJG) used in this study is archived on Zenodo at (https://doi.org/10.5281/zenodo.16908049, Jia et al., 2025). All datasets used for model setup and forcing are openly available. The sources of the corresponding datasets are out lined in detail in Sect. 2.

**Author contribution**

ZTJ and YSF conceived and designed the study. ZTJ wrote the model coupling code, designed and executed the model experiments, and wrote the manuscript with input and supervision from YSF. JT provided technical support and guidance.



All coauthors contributed to the analysis and interpretation of results, discussed the findings, and reviewed and edited the manuscript to finalize it for submission.

**Competing interests**

The authors declare that they have no conflict of interest.

**Acknowledgements**

The authors thank the ParFlow and LPJ-GUESS development teams for providing the modelling tools that made this study possible.

**Finance support**

This research was supported by the Key Program of the National Natural Science Foundation of China (Grant No. 42430504), the National Key Research and Development Program of China (Grant No. 2023YFF0805604), the Fundamental Research Funds for the Central Universities (2243300004), and the 111 Project (Grant No. B18006).

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
