# Peer review of "Advancing Ecohydrological Modelling: Coupling LPJ-GUESS with ParFlow for Integrated Vegetation and Surface-Subsurface Hydrology Simulations"

_EGUsphere, 2025_

## Author Comment (AC5)

**Response to Reviewer #1:**

In this manuscript the authors propose a robust coupled model for integrated vegetation and surface-subsurface water flow simulations. This work is very valuable for the egusphere community. It is mostly clear but would needs some additional clarification, illustrations and discussions to improve its readability and repeatability.

**Response:** Sincere thanks to the reviewer for the kind consideration and constructive comments on our manuscript. We have carefully addressed all the comments and revised the manuscript accordingly. A point-by-point response is provided below (in blue), and all corresponding changes have been updated in the revised manuscript. We hope these changes will strengthen our manuscript.

**Main comments:**

[Comment 1] Parflow equations and annotations need some clarifications. Describing each equation briefly by one sentence summarizing what it does / means and how they are linked/solved with respect to each other would bring some clarity to the reader (there is already some attempt, but it is still a bit confusing). Time discretization shall be introduced and justified for ParFlow and LPJ-GUESS.

**Response:** Revised. Thanks to the reviewer for the insightful suggestion; Following your recommendation, we have made several improvements: redundant equations have been removed and key equations highlighted; each ParFlow-related equation now includes a brief description summarizing its meaning and role; the relationships among the equations and their solution methods have been clarified; all variables are clearly defined; and time discretization for both ParFlow and LPJ-GUESS has been introduced and justified. The corresponding revisions are detailed in the Methods section (lines 127-155).

[Comment 2] The value of input parameters should be included in the manuscript. Providing references from where it was sourced is great but not sufficient, given that there is no calibration of those parameters. Consider to illustrate them with some figures: landcover map, topography and river map, annual rainfall map and timeseries at one location, hydrogeological model vertical cross section (hydrostratigraphic units, heterogenous property fields), soil data property maps per layer or vertical cross-section. For homogeneous properties inside one layer or a hydro-stratigraphic unit, a table summarizing the parameter values should be provided.

**Response:** Revised. Thanks to the reviewer for the practical suggestion. We have added several supplementary datasets, including the topographic DEM, long-term mean landuse distribution, depth to bedrock (DTB), long-term mean potential recharge (P-ET), as

well as the layered hydrogeological units and soil property maps. Detailed classification descriptions are also provided in the supplementary materials (Fig. S1-S3).

In our model, the parameterization of soil layers and the bedrock layer follows the configuration in Table S1 of the supplementary materials of Maxwell and Condon (2016) (Table S1).

Maxwell, R. M. and Condon, L. E.: Connections between groundwater flow and transpiration partitioning, Science, 353, 377-380, https://doi.org/10.1126/science.aaf7891, 2016.

[Comment 3] The discussion should be separated from the results section. A more thorough discussion should be written to acknowledge the limitations of the current coupled model and suggest possible modelling improvements on both aspects of the coupled model (Vegetation-Land Surface aspect and Hydro-geo-logical aspect). Given the not so good results of Water Table Depth, calibration of hydrogeological parameters should be discussed with respect to the studied Danube basin or other locations. There is also a river network pattern for ET and SM produced by the coupled model, that are not present in the 'reference' data; that should also be discussed.

**Response:** Added. Thanks to the reviewer for the constructive suggestion. We have separated the results and discussion section and supplied the more discussion section. As per your instructions, we have acknowledged the limitations of the current coupled model within the discussion and proposed potential modelling improvements have been proposed for two aspects of the coupled model: the vegetation-surface interaction and the hydrogeological component. Discussions on calibrating hydrogeological parameters have been incorporated.

Furthermore, we added a discussion on why the coupled model produces river-network-like patterns in ET and SM, which are not present in the reference datasets. This arises from ParFlow's assumption of fully saturated river channel grid cells, enhancing SM and ET along river grids. While the GLEAM4 dataset accounts for water body ET, but its coarse land cover resolution does not resolve smaller rivers in the Danube basin. This highlights the PF-LPJG model's ability to capture fine-scale riverine hydrological processes.

These points are now included in the revised manuscript (sections 3 and 4) and clearly distinguish model strengths, limitations, and potential improvements.

**Detailed comments:**

[Comment 3] Abstract is clear.

**Response:** Thanks to the reviewer for the positive evaluation of the abstract.

[Comment 4] Introduction is clear.

**Response:** Thanks to the reviewer for the positive assessment of the introduction.

**2.2 Parflow**

[Comment 5] Equation (1) and lines 123-124: is it phi\_p (not defined after) or psi\_p in equation 1? Should it be  $q_s(x)$  in equation 1 instead of  $q_e(x)$ ?

**Response:** Corrected, the  $\phi_p$  is  $\psi_p$  in equation (1), we have corrected this equation in my article. Yes, qs(x) in equation 1 should be instead of  $q_e(x)$ .

[Comment 6] Line 126, do you mean the boundary conditions q\_bc?

**Response:** Corrected, we intended to refer to the Neumann-type boundary condition  $q_e(x)$  and this has now been clarified.

[Comment 7] Lines 129-130: are (1) and (2) not the same thing? psi\_p = psi\_s seems related to the sentence after.

**Response:** Corrected, points (1) and (2) referred to the same condition, and this redundancy has been removed.

[Comment 8] Line 130: define psi s here.

**Response:** Corrected, we have reordered the equations and introduced the definition of  $\psi_s$  at its first occurrence.

[Comment 9] Equations 4 and 5: what is q r?

**Response:** Corrected, all  $q_r(x)$  terms were intended to represent  $q_s(x)$ , we have corrected these and now consistently use  $q_s(x)$  as the general source-sink term.

**2.3 Coupling model approach**

[Comment 10] Timestep discretization needs to be introduced in 2.2 to clarify the articulation of the coupled ParFlow and LPJ-GUESS models; it seems clear that there is a daily time scale interaction, but time discretization could potentially be different between the solvers (finer different discretization).

**Response:** Clarified. Thanks to the reviewer for the professional suggestions, we added an explanation of the timestep discretization in lines 119-120, describing how ParFlow and LPJ-GUESS are coupled on a daily timescale and how differences in internal solver timesteps are handled, please see lines 172-174.

[Comment 11] Figure 1: for consistency, keep the same left-right ordering of soil moisture / precipitation

**Response:** Corrected. Thanks to the reviewer for the suggestion. The left-right ordering

has been corrected to make consistent in Figure 1.

**2.4 Data sets**

[Comment 12] Line 172: "in a lot of research" seems superfluous, remove it.

**Response:** Corrected. Thanks to the reviewer for the suggestion. The phrase has been removed for conciseness.

[Comment 13] Lines 177-179: data at different resolution? Please check and clarify the resolution used for each data-type. What is "u-component of wind"? What is the CDS daily aggregation method?

**Response:** Clarified. Thanks to the reviewer for pointing this out. We have standardized and clarified the resolutions of all datasets used, as detailed in the revised manuscript. The u-component of wind refers to the east-west component of horizontal wind velocity. The CDS daily aggregation method has also been explained in the revised manuscript, please see lines 195-196.

[Comment 14] How many river flow observation points from the Danube River Basin are used? Where are they located?

**Response:** Thanks to the reviewer for the helpful feedback, we have updated the figures as required. Seven GRDC river flow observation points were used in this study, their geographical locations have been added to Figure 3 in the revised manuscript.

[Comment 15] Can you explain why the GLEAM data can be used as a reference as it is the result of a model?

Same justification needed for the ESA CCI-SM product.

**Response:** Thanks to the reviewer for the insightful feedback. Although GLEAM v4 is model-derived, it is an observation-driven evapotranspiration dataset that integrates satellite-based remote sensing (e.g., surface soil moisture, vegetation optical depth) with physically based formulations such as the Penman-Monteith equation. GLEAM has undergone extensive validation against flux-tower and in situ observations and has been widely used as a benchmark in hydrological and land-surface studies.

Similarly, the ESA CCI-SM product is generated through the harmonization and fusion of multi-sensor microwave satellite observations. It is primarily observation-based and has been validated globally against ground-based measurements. Owing to their long temporal coverage, global consistency, and demonstrated reliability, both datasets are broadly used as reference datasets for model evaluation. These justifications have been added to the revised manuscript, please see lines 255-257.

[Comment 16] How many in-situ water table depth observation points from the Danube Basin are used? Where are they located?

**Response:** Thanks to the reviewer for the insightful question. The dataset of all 48 insitu water table depth observation points only contains the annual average water table depth values from the Fan's paper (Fan et al. (2013)). All available observation points within the Danube Basin are located in the downstream region. These clarifications have been added to the revised manuscript (lines 479-480).

[Comment 17] 2.5 Line 241:by "stabilizes less than 1 %" do you mean "stabilizes, with fluctuations less than 1 %"?

**Response:** Clarified. Thanks to the reviewer for pointing this out. Yes, we mean "stabilizes, with fluctuations less than 1%". As noted in lines 278-279 of the revised manuscript, the model is run until the change in groundwater storage stabilizes, with fluctuations less than 1% of the potential recharge (P-ET).

**3.1 Streamflow**

[Comment 18] Global and local results: the boxplots could also be presented at the gauging station level to quantify the performance as a function of basin size. A sketch showing the relationship of the 7 considered basins (sub-basin of the main one), would help the reader understand their relationship, rather than guessing it.

**Response:** Corrected. Thanks to the reviewer for the revision suggestions. We have revised Figure 2 and 3 according the reviewer's suggestion, changing the Figure 2 to present boxplots at each gauging station, allowing performance evaluation as a function of basin size. Modify the left-hand sub-plot of Figure 3 to indicate the spatial extent of the seven sub-basins within the study area and the locations of the corresponding monitoring stations.

**3.2 ET**

**[Comment 18]** Figure 4: subtitles of 1st and 2nd row too long, make it hard to read, maybe add Annual ET Difference to the colour-bar legend/label. The colormap to show the difference is not great (subplots d to f): use a seismic or bwr (blue white red) colormap centred around 0 such that white colour denotes no change, blue colours negative difference and red colours positive difference. Subplots a to c: use a colourblind linear colormap.

Subplots a-f: missing scale and North. Subplot (g): missing unit on the x axis

Response: Corrected. Thanks to the reviewer for pointing this out, we have made

corresponding adjustments. We shortened the subtitles of the first and second rows and incorporated "Annual ET" Difference directly into the color-bar labels. For subplots (d)-(f), we replaced the previous colormap with a seismic (RdBu\_r) colormap centered at zero (white = no change, blue = negative difference, red = positive difference). Subplots (a)-(c) now use a color-blind-friendly linear colormap "viridis". A scale bar and a North arrow have been added to all spatial subplots, and units have been added to the x-axis of subplot (g).

**3.3 SM**

[Comment 19] Figure 5: what does SWC stands for? Subplot (c): use a seismic or bwr (blue white red) colormap centred around 0 such that white colour denotes no change, blue colours negative difference and red colours positive difference. Subplots a,b and d: use a colourblind linear colormap.

**Response:** Corrected. Thanks to the reviewer for the careful reminder and have made corresponding adjustments. SWC stands for "soil water content". We unified all terminology to "SM" (soil moisture) throughout the manuscript for consistency. Subplots (a) and (b) have been updated to a color-blind-friendly linear colormap "viridis r," and subplot (c) now uses a seismic (RdBu r) colormap centered at zero.

For subplots (d), we attempted to update subplot (d) using a standard colorblind-friendly colormap; however, this reduced the perceptibility of subtle differences and could introduce visual artifacts in transition zones, potentially compromising interpretability. To preserve both scientific accuracy and visual clarity, we have updated subplot (d) by adding colors to the original palette as a basis, making it compliant with colorblind-friendly requirements and thereby ensuring accessibility and improved readability.

[Comment 20] Figure 6: not sure how the RMSE and Sperman rho CDFs were calculated, for how many subsamples? How were the subsamples selected?

**Response:** Thanks to the reviewer for the careful and very important reminder. The conditional distribution functions (CDFs) for the root mean square error (RMSE) and Spearman's correlation coefficient were computed using monthly observed and simulated values from 1980 to 2018 across all grid cells within the Danube River Basin. Thus, each grid cell-month pair constitutes one sample, and all grid cells in the basin were included. We have added the detail in line 443.

**3.4 Water Table Depth**

[Comment 21] The CDF error from the PF-LPJG is smaller than from the work of Fan

et al. (2013) but it does not seem close to the real observations and strongly biased (shifted cdf). That should be acknowledge and some plausible explanation given. Figure 8c should be compared to real observations at least once; it is not possible to observe to identify spring or other seasons on the graph as year graduations are too small, maybe zoom over a smaller time range to support this statement.

**Response:** Thanks to the reviewer for the professional suggestion. We have acknowledged this point and give a plausible explanation in the paper. We can't compare with real observations because of the dataset of all in-situ water table depth observation points contains only annual mean water table depth values. Furthermore, we have added in the Supplementary Material a plot of the monthly mean groundwater table depth (WTD) averaged over the period 1980-2018, based on our simulated results, which facilitates the identification of seasonal patterns (Fig. S4). For instance, groundwater rises during spring and summer when there is recharge and declines in winter when there is no recharge. Please see the lines 480-493.

All in-situ wells are located in the lower reaches of the Danube River, where groundwater depths typically range from 0 to 40 m. In this area, soil moisture oversaturation leads PF-LPJG to simulate very shallow water table depths (0-1 m) at certain locations. In regions with relatively deeper groundwater, the modelled WTD ranges from 0 to 20 m, showing substantially better agreement with observational data than the Fan et al. (2013) dataset, which systematically overestimates WTD (typically exceeding 40 m) in these lowland environments. Overall, PF-LPJG more accurately reproduces both the magnitude and spatial variability of groundwater depths across the basin.

**3.5 E T partitioning**

[Comment 22] Figure 8a is hard to read; create another subplot to separate the evaporation series from the transpiration series. Maybe an additional plot of the residuals as time series would facilitate the interpretation of these results.

**Response:** Revised, thanks to the reviewer for the helpful suggestion. In the revised manuscript, we have created an additional subplot to separate the evaporation and transpiration series, improving the readability of Figure 8a. We did not add a residual time series plot, as it does not provide further insight into the model comparison, the revised Figure 8 already clearly illustrates the differences between the simulations.

**Supplementary information**

**Figure S1.** The basin characteristics used in the model: (a) Digital Elevation Model (DEM) processed by PriorityFlow, (b) Annual mean landuse distribution, (c) Thickness of unconsolidated bedrock, (d) Annual mean net water input (P-ET) used as recharge flux. Land use from IGBP Global Vegetation Classification (1-17): 1. Evergreen Needleleaf Forests; 4. Deciduous Broadleaf Forests; 5. Mixed Forests; 8. Woody Savannas; 9. Savannas; 10. Grasslands; 11. Permanent Wetlands; 12. Croplands; 13. Urban; 14. Cropland/Natural Vegetation Mosaic 15. Snow and Ice.

**Figure S2.** Classification of soil properties: (1) sand, (2) loamy sand, (3) sandy loam, (4) silt loam, (5) silt, (6) loam, (7) sandy clay loam, (8) silty clay loam, (9) clay loam, (10) sandy clay, (11) silty clay, (12) clay. Categories with few grid cells have been displayed using the same color.

**Figure S3.** Classification of bedrock layers: (19) bedrock 1, (20) bedrock 2, (21) f.g. sil. sedimentary, (22) sil. sedimentary, (23) crystalline, (24) f.g. unconsolidated, (25) unconsolidated, (26) c.g. sil sedimentary, (27) carbonate. Note that f.g., sil., and c.g. represent fine-grained, siliciclastic sedimentary, and coarse-grained, respectively. Hydraulic conductivity increases with increasing layer number.

**Figure S4.** Monthly mean groundwater table depth (WTD) averaged over the period 1980-2018, based on simulated results.

Table S1. Parameters of Soil and Bedrock Layers

| Class            | Unit Indicator | Classification        | Ks (m/h) | porosity [-] | sres [-] | alpha (1/m) | n [-] |
|------------------|----------------|-----------------------|----------|--------------|----------|-------------|-------|
|                  | 1              | Sand                  | 2.69E-01 | 0.38         | 0.14     | 3.55        | 4.16  |
|                  | 2              | Laomy Sand            | 4.36E-02 | 0.39         | 1.26     | 3.47        | 2.74  |
|                  | 3              | Sandy Loam            | 1.58E-02 | 0.39         | 0.10     | 2.69        | 2.45  |
|                  | 4              | Silt Loam             | 7.58E-03 | 0.44         | 0.15     | 0.50        | 2.66  |
|                  | 5              | Silt                  | 1.82E-02 | 0.49         | 0.10     | 0.66        | 2.66  |
| Soil             | 6              | Loam                  | 5.01E-03 | 0.40         | 0.15     | 1.12        | 2.48  |
| Units            | 7              | Sandy clay loam       | 5.49E-03 | 0.38         | 0.16     | 2.09        | 2.32  |
|                  | 8              | Silty clay loam       | 4.68E-03 | 0.48         | 0.19     | 0.83        | 2.51  |
|                  | 9              | Clay loam             | 3.39E-03 | 0.44         | 0.18     | 1.58        | 2.41  |
|                  | 10             | Sandy clay            | 4.78E-03 | 0.39         | 0.30     | 3.31        | 2.20  |
|                  | 11             | Silty clay            | 3.98E-03 | 0.48         | 0.23     | 1.62        | 2.32  |
|                  | 12             | Clay                  | 6.16E-03 | 0.46         | 0.21     | 1.51        | 2.26  |
| Bedrock
Units | 19             | Bedrock 1             | 5.00E-03 | 0.33         | 0.001    | 1.00        | 3.00  |
|                  | 20             | Bedrock 2             | 1.00E-02 | 0.33         | 0.001    | 1.00        | 3.00  |
|                  | 21             | f.g. sil. Sedimentary | 2.00E-02 | 0.30         | 0.001    | 1.00        | 3.00  |
|                  | 22             | sil. Sedementary      | 3.00E-02 | 0.30         | 0.001    | 1.00        | 3.00  |
|                  | 23             | crystalline           | 4.00E-02 | 0.10         | 0.001    | 1.00        | 3.00  |
|                  | 24             | f.g. unconsolidated   | 5.00E-02 | 0.30         | 0.001    | 1.00        | 3.00  |
|                  | 25             | unconsolidated        | 6.00E-02 | 0.30         | 0.001    | 1.00        | 3.00  |
|                  | 26             | c.g. sil sedimentary  | 8.00E-02 | 0.30         | 0.001    | 1.00        | 3.00  |
|                  | 27             | carbonate             | 1.00E-01 | 0.10         | 0.001    | 1.00        | 3.00  |

---

## Author Comment (AC6)

**Response to Reviewer #2:**

The authors present a study coupling the 3D variably saturated groundwater surface water model with the ecosystems model LPJ-GUESS. The rational is that applying coupling groundwater processes are improved which in turn improve the representation of land surface and ecosystems' processes. The coupled model is applied over the Danube basin at the climate time scale. A comprehensive evaluation of the model with observations is performed that shows that indeed the coupled model is able to represent elevation processes much better even without calibration.

This is an interesting and timely study. The terrestrial system constitutes a continuum with respect to the coupled water, energy and matter cycles. It models it should be treated as such, in order to arrive at interpretable results and allow extrapolation into future climates. It is curious that the comparison to observations results in excellent statistical metrics especially considering the no calibration was applied. This has been observed in previous coupled modeling studies (also with ParFlow). It would be interesting to hear the authors' opinion why this is.

Below I provide general and specific comments and suggestions. The study can be published following moderate to major revisions.

Thank you for your positive evaluation. We will carry out major revisions and improve the paper comprehensively.

Response: We are most grateful to the reviewer for the constructive comments and encouraging assessment of our work, acknowledging it as an interesting and timely study, and noting that our coupled model demonstrates excellent simulation of hydrological variables without requiring calibration. This characteristic reflects the physics-based nature of the coupled model, where hydrological and ecological fluxes are explicitly represented and constrained by observed boundary conditions and physical parameters. The river network within the catchment evolves organically through the evolution of infiltrating surface water and groundwater flows, rather than through empirical adjustments. Considering the complexity and physical realism of the input data, the results were obtained without calibration. We will carry out major revisions following your suggestion and improve the paper comprehensively.

**Introduction**

[Comment 1] Given the plethora of work that as been done with physics-based groundwater flow models, in particular with ParFlow with reagard to coupling with land surface and also atmospheric processess, the introduction is not only incomplete, but misses to contextualize the novelty of the presented study. The gap mentioned on

line 70 has been bridged in many other previous studies.

Response: Revised, thanks to the reviewer for pointing out the need to better contextualize the novelty of our study. Following the reviewer's suggestion, we have revised the introduction to include details on ParFlow's current coupling status and remaining process gaps. In particular, we added discussions of previous coupling efforts of ParFlow with land surface and atmospheric models such as WRF, CLM, LIS, and Noah-MP, and clarified the remaining research gaps. Specifically, most prior studies focused on coupling ParFlow with land surface processes but employed static vegetation parameters, which do not adequately represent dynamic vegetation growth and ecosystem processes (e.g., vegetation mortality, interspecific competition, phenology, and carbon-nitrogen cycling).

The novelty of our study lies in: Coupling groundwater-soil water processes with dynamic vegetation and ecosystem processes through bidirectional feedback; Treating vegetation as an active participant providing feedback to the hydrological system, rather than as a passive recipient of water flux; Capturing nonlinear feedbacks of vegetation dynamics on soil moisture, groundwater, and transpiration.

These additions are now reflected in lines 67-70 of the revised manuscript.

[Comment 2] ParFlow: Which type of shallow water equation was used, kinematic of diffusive?

**Response:** The overland shallow flow in ParFlow is computed using the Overland Kinematic Wave equation.

**Coupling modeling approach**

[Comment 3] Is the coupling directed at the existing coupling of ParFlow with e.g. the Common Land Model? Is daily coupling sufficient? What's the total depth of the ParFlow model?

**Response:** Thanks to the reviewer for these insightful questions. Our responses are provided below:

- 1. Coupling framework: In our coupled model, the coupling is implemented directly with ParFlow 3.13, which does not directly couple with ParFlow-CLM but only with ParFlow itself. we have effectively replace the typical ParFlow-CLM coupling to couple Parflow with a new ecosystem model, (dynamic vegetation process model LPJ-GUESS. This allows the coupled system to explicitly simulate both hydrological and ecological processes (updated in the revised paper, please see line 159).
- 2. Temporal discretization: ParFlow operates on an hourly timescale to capture fast

hydrological processes, while LPJ-GUESS updates daily to reflect vegetation growth processes. In practice, LPJ-GUESS reads ParFlow outputs every 24 timesteps (1 day) to compute daily water fluxes. These daily P-ET values are disaggregated into hourly values according to diurnal patterns and passed back to ParFlow for the next simulation day. This strategy preserves hydrological fidelity while maintaining realistic ecological dynamics (updated in the revised paper, please see lines 173-175).

3. Model depth: The total depth of the ParFlow model in the Danube River Basin is set at 92 meters below the surface (depending on the specified bedrock depth of the Danube River Basin) and divided into 10 layers with variable thicknesses (from bottom to top: 50, 25, 10, 10, 10, 5, 1, 0.6, 0.3, 0.1 m). Layer thicknesses can be adjusted according to the hydrogeology and research objectives on the specific study area (updated in the revised paper, please see lines 255-257).

[Comment 4] Soil data: Why not use Soil Grids instead?

**Response:** We thanks to the reviewer for this helpful suggestion. We have revised the description of the soil dataset for greater clarity and consistency with the subsurfacedata explanation. Although Soil Grids provides global, high-resolution soil information, LPJ-GUESS requires soil input that is directly compatible with the WISE soil texture dataset, which is used in its parameterization. In addition, the WISE-derived soil texture fractions better reflect the spatial variability of soil properties. Therefore, we used 10-km soil grids derived from the WISE database, where each grid contains sand, silt, and clay percentage fractions. These revisions have been updated in the revised paper, please see lines 197-205.

**Subsurface data:**

[Comment 5] Is soil data not subsurface data? For each hydrogeologic unit an indicator is required not an indicator file. To which depth was he subsurface represented in the model.

**Response:** We thanks to the reviewer for raising these important points, and we have substantially revised our description of the subsurface data. Soil data alone does not fully represent subsurface characteristics. In the coupled model, subsurface properties are assigned using a 3-D indicator array, which specifies the hydrogeologic unit for each grid cell and depth level.

Soil layers (0-2 m; codes s1-s12): Derived from WISE soil texture fractions (sand, silt, clay). Twelve soil classes were defined using the soil texture classification code. Each soil class is linked to predefined hydraulic parameters, including hydraulic conductivity, porosity, and relative permeability curves.

Bedrock layers (2-92 m; codes g19-g28): Derived from GLHYMPS v1.0 permeability data and grouped into classes based on hydraulic-conductivity distributions.

Each unit class has a set of hydraulic properties. The 10 vertical subsurface layers (soil + bedrock) in the model are therefore defined through these indicator codes, which are read by the model to assign parameter values. All relevant revisions appear in lines 197-215.

[Comment 6] 54: These references to not show influence of groundwater on boundary layer development. An example is Rahman, M., Sulis, M., Kollet, S.J., 2015. The subsurface—land surface—atmosphere connection under convective conditions. Advances in Water Resources 83, 240–249. https://doi.org/10.1016/j.advwatres.2015.06.003.

**Response:** Revised, thanks to the reviewer for the professional comment. The suggested reference has been added to the revised manuscript (please see line 53).

[Comment 7] 116: It is not the jacobian matrix that is solved but a system of equations.

**Response:** Corrected, thanks to the reviewer for pointing this issue out. Following the reviewer's suggestion, we have corrected the sentence to clarify that the Newton-Krylov framework in ParFlow solves the Jacobian-based linear system using GMRES, rather than directly solving the Jacobian matrix itself (please see lines 120-121).

[Comment 8] 216: The slope was calibrated? Perhaps calculated.

**Response:** Yes, we appreciate the reviewer's insightful comment. As clarified in lines 239-241 of the manuscript, the slope variable is derived from the priority flow algorithm, which calculates slope values in both the x and y directions for each grid cell based on regional topography and river network data. The algorithm uses a D4 flow direction scheme, and the resulting drainage area was validated against the 0.1° drainage area dataset from the IHU publication (Eilander et al., 2021). Details of the terrain preprocessing and algorithm implementation are provided in the priority flow algorithm and topographic preprocessing references (Zhang et al., 2021).

Eilander, D., van Verseveld, W., Yamazaki, D., Weerts, A., Winsemius, H. C., and Ward, P. J.: A hydrography upscaling method for scale-invariant parametrization of distributed hydrological models, Hydrol. Earth Syst. Sci., 25, 5287-5313, https://doi.org/10.5194/hess-25-5287-2021, 2021. Zhang, J., Condon, L. E., Tran, H., and Maxwell, R. M.: A national topographic dataset for hydrological modelling over the contiguous United States, Earth Syst. Sci. Data, 13, 3263-3279, https://doi.org/10.5194/essd-13-3263-2021, 2021.

[Comment 9] Equation 6 is redundant.

**Response:** Corrected. Thanks to the reviewer for the professional suggestion, the redundant Equation (6) has been removed in the revised manuscript.

**Results and Discussion**

[Comment 10] 323: ParFlow-CLM

In order to show uncertainty in ET observations, it would be nice to include more than one product in the analyses. In figure 4h, please include the results from LPJ-GUESS.

**Response:** Added, thanks to the reviewer for the constructive suggestion, and have revised the manuscript accordingly. We have added a comparison with the LPJ-GUESS seasonal variability in Figure 4h.

Regarding the use of additional ET products, we carefully considered incorporating MODIS16 as an extra observational benchmark. However, our evaluation shows that MODIS16 systematically produces higher ET values (typically 600-800 mm yr-1 across most parts of the basin) and exhibits spatial patterns that differ substantially from GLEAM, LPJ-GUESS, and PF-LPJG. Because these differences largely reflect known structural biases in MODIS16 rather than model-observation discrepancies, including MODIS16 would complicate interpretation without providing a consistent reference for model evaluation.

Given that GLEAM has been extensively validated with flux tower observations and is widely recognized as one of the most reliable global ET datasets, we use it as our primary benchmark in this study. Nevertheless, we acknowledge that MODIS16 may offer complementary insights for future multi-product intercomparison analyses.

[Comment 11] Interestingly, ParFlow simulates too shallow water tables. What are the reaons? Coarse resolution, hydrogeologic heterogeneity, etc.? Perhaps it's worthwhile considering anisotropy in the hydraulic conductivity, where Kxx, Kyy > Kzz.

**Response:** Thanks to the reviewer for the professional suggestion. In our coupled simulations, the relatively coarse 10-km model resolution (ensure computational feasibility) tends to smooth topographic gradients and river channel geometry. This smoothing promotes soil moisture accumulation in low-slope areas, leading to locally excessive saturation and consequently shallower simulated water table depths compared with observations.

With respect to hydraulic conductivity, PF-LPJG already applies an anisotropic scheme in which horizontal conductivity ( $K_x = K_y$ ) exceeds vertical conductivity ( $K_z$ ), with  $K_z$  typically set to  $0.01 \times K_{x,y}$  to represent flow-restricting layers. In future work, we plan to explore increasing  $K_x$  and  $K_y$  to better represent the lateral spreading of groundwater

and improve the simulation of water table depth in low-gradient terrains. These limitations and potential improvements have been incorporated into the revised manuscript (please see section 4.3).

**Insight**

[Comment 12] Can the authors provide some insight on the computational or HPC aspect of the simulations since the computational burden of coupling ParFlow with LPJ GUESS increases considerably in the context of decadal scale simulations.

**Response:** Thanks to the reviewer for raising this important point, which is indeed a common concern among users of high-resolution coupled models. In our current setup, the coupled PF-LPJ-GUESS model runs at a resolution of 10 km using 80 + 100 CPU cores, requiring approximately 6-7 hours of computation per simulated year. Both ParFlow and LPJ-GUESS support large-scale parallel computation on HPC systems. In future work, we plan to adopt the GPU-accelerated version of ParFlow, which combines MPI and GPU parallelization, potentially improving computational efficiency by a factor of 2-3 compared to the current setup.

[Comment 13] What does the coupling of ParFlow with LPJ-GUESS do wrt to non-linear interactions and the boundary value problem of the terrestrial hydrologic cycle and how it is generally treated in land surface models? What are the implications for Earth system modeling?

Response: Thanks to the reviewer for these thought-provoking questions. In the ParFlow-LPJ-GUESS configuration, LPJ-GUESS dynamically simulates vegetation growth, competition, and carbon cycling in response to hydrological conditions, while ParFlow resolves the three-dimensional variably saturated flow that governs soil moisture and groundwater dynamics. Through this coupling, the terrestrial hydrologic cycle is formulated as a dynamic boundary value problem (BVP), where both upper and lower boundaries evolve interactively in time, rather than being treated as static parameters or externally prescribed fluxes. Such a formulation allows nonlinear feedback between vegetation physiology (e.g., transpiration, rooting depth, leaf area) and hydrological states (e.g., soil moisture, groundwater level) to emerge explicitly.

In contrast, the ParFlow-CLM coupling mainly emphasizes the exchanges of surface energy and water fluxes associated with land-atmosphere interactions. The coupling of ParFlow-LPJ-GUESS focuses on representing the two-way interactions between vegetation dynamics and subsurface-surface hydrological processes. Although both coupling frameworks exchange similar variables, the processes that generate and respond to these fluxes differ fundamentally.

This integrated approach provides a more realistic representation of vegetation—water interactions, especially across arid and humid regions, and improves the physical consistency of land carbon-water coupling. Consequently, it enhances the representation of terrestrial processes in Earth System Models (ESMs) and strengthens the simulation of long-term (decadal) water-carbon-energy feedbacks. The discussion on these questions has been included in the revised manuscript (sections 3 and 4).